# Antisymmetric Lamb Wave Simulation Study Based on Electromagnetic Acoustic Transducer with Periodic Permanent Magnets

**DOI:** 10.3390/s23167117

**Published:** 2023-08-11

**Authors:** Lianren Du, Ruizhen Gao, Xiaojuan Jia

**Affiliations:** College of Mechanical and Equipment Engineering, Hebei University of Engineering, Handan 056038, China; dulianren99853@163.com (L.D.); gaoruizhen@hebeu.edu.cn (R.G.)

**Keywords:** Lamb wave, electromagnetic acoustic transducer, periodic permanent magnet, mode, total displacement amplitude ratio

## Abstract

Due to its multi-mode and dispersion characteristics, Lamb waves cause interference to signal processing, which profoundly limits their application in nondestructive testing. To resolve this issue, firstly, based on the traditional EMAT, a horizontal polarization periodic permanent magnet electromagnetic acoustic transducer (HP-PPM-EMAT) was proposed. A 2-D finite element model was then developed to compare magnetic flux density, Lorentz force, and signal strength between the traditional EMAT and the HP-PPM-EMAT. The simulation results show that the HP-PPM-EMAT enhances the A0 mode Lamb wave (A0 wave) and suppresses the S0 mode Lamb wave (S0 wave). Finally, the influence of structural parameters of the HP-PPM-EMAT on the total displacement amplitude ratio of A0 and S0 was investigated using orthogonal test theory, and the width of magnet units was improved based on the orthogonal test. The results show that the total displacement amplitude ratio of A0 to S0 of the improved HP-PPM-EMAT can be improved by a factor of 7.74 compared with that of the traditional Lamb wave EMAT, which can produce higher-purity A0 mode Lamb waves.

## 1. Introduction

Commonly, acoustic testing transducers in industry include electromagnetic acoustic transducers (EMATs) and piezoelectric transducers (PZTs). Among them, the EMAT is a non-contact acoustic transducer that excites and receives acoustic waves by means of electromagnetic coupling. Unlike PZTs, it does not require coupling agent in the detection process, which significantly improves the detection efficiency. In addition, EMATs can work in particular environments such as high temperatures, high speeds, and isolation layers [1,2,3,4]. Through different combinations of coils and magnets, EMATs can generate and receive various types of ultrasonic waves in metal samples, such as guided waves, shear waves, longitudinal waves, and surface waves.

Ultrasonic Lamb waves are guided waves with the advantages of long propagation distance and low attenuation [5]. Therefore, Lamb waves are widely used in nondestructive testing and evaluation of plate structures [6,7,8]. According to the symmetry of particle motion relative to the midplane of the plate, Lamb waves are divided into symmetric modes (S0, S1, S2, …) and antisymmetric modes (A0, A1, A2, …). The symbol n in Sn and An represent modules, and S0 and A0 represent the lowest-order symmetric and asymmetric modes, respectively. Multi-mode and dispersion phenomena exist in Lamb waves under the same excitation condition, resulting in confusion and overlap of the received echo signals of all modes, which seriously affects the detection accuracy and echo signal analysis [9]. Therefore, it is essential to excite a single-mode Lamb wave for later detection and signal analysis.

However, compared with the PZT, the EMAT has poor energy conversion efficiency and is prone to noise interference. Meanwhile, the signal analysis of the Lamb wave is complicated due to its multi-mode and dispersion. Therefore, improving the low energy exchange efficiency of the EMAT and the multi-mode of Lamb waves has become a research focus. Permanent magnets, as an important component of EMATs, mainly provide static magnetic fields. To enhance the signal intensity, increasing the flux density of the static magnetic field is a direct and effective method. Zhang Xu et al. [10] proposed a flux-focused EMAT composed of radial flux-focusing magnets, which provided a strong horizontal radial magnetic field and vertical axial magnetic field, significantly improving the amplitude of the studied wave. By adding a magnetic concentrator to the bottom of the magnet to guide and concentrate the static magnetic field of the permanent magnet, Liu et al. [11] removed the unwanted A0 mode and then improved the signal purity and strength.

In addition, the researchers put forward a series of solutions from the design of the permanent magnet and coil structure and the selection of the excitation signal. GUO et al. [12] developed an EMAT consisting of a racetrack coil and periodic permanent magnets. Then, they constructed two models according to the position of the coil to increase the signal intensity of S0 and A0, respectively. Liu et al. [13] proposed a dual-EMAT excitation configuration to generate a single-mode Lamb wave, reducing subsequent signal processing difficulty and providing a theoretical basis for practical defect detection. Wilcox et al. [14] proposed an omnidirectional transducer that uses a spiral coil and cylindrical permanent magnet to excite the S0 mode. Yang et al. [15] proposed a periodic magnet EMAT configuration, which can significantly enhance the amplitude of S0 mode by using periodically arranged magnets to increase the magnetic flux density of the local magnetic field. Sun et al. [16] proposed an EMAT in which the magnet consisted of a cylindrical magnet and a ring magnet around the outside of the cylindrical one, which eliminated the multi-mode phenomenon and improved the purity of the A0 wave. Jiao et al. [17] developed a single A0 wave PZT array, which was applied to detect defects in a wide range of plate structures. Zhai et al. [18] combined the main design parameters of the EMAT with the Lamb wave characteristic equation to establish a new electromagnetic ultrasonic Lamb wave excitation equation and dispersion curve and then suppressed the influence of multi-mode and dispersion on the signal by selecting the appropriate excitation working point. Li et al. [19] derived the analytical expression of the modal expansion coefficient of the Lamb wave by using the modal expansion method. Under the given geometric parameters, the mode of the Lamb wave can be effectively adjusted by selecting the appropriate EMAT driving frequency.

To improve the excitation performance of Lamb wave EMAT, Kang et al. [20] proposed an optimal design method that combines the “two-intersection-points” method and the “zero-slope” criterion with the orthogonal test method. Although the signal intensity is improved by this method, the horizontal magnetic flux density generated by vertically polarized permanent magnets will affect the coil on both sides, which is not conducive to generating single-mode Lamb wave. Liu et al. [21] reduced the eddy current on the surface of the permanent magnet by adding super-microcrystals between the permanent magnet and the coils, which improved the distribution of the static magnetic field, thus achieving the purpose of enhancing the intensity of the ultrasonic signal, but at the same time, enhanced the signal amplitude of the symmetric mode and the antisymmetric mode. Kubrusly et al. [22] investigated the dual emission and dual reception technique of PPM EMATs combined with SH-guided waves. Compared with the electromagnetic ultrasonic transducer consisting of a single horizontally polarized permanent magnet and wire array coil [23], the technique can clearly separate the symmetric mode from the antisymmetric mode in the processed received signal. Zhang et al. [24] proposed a pairing structure of curved coil EMAT, in which two horizontally polarized magnets were placed on the tested part’s upper and lower surfaces, respectively. The horizontal Lorentz forces generated by the pairing EMAT cancel each other and effectively control the generation of the A0 mode. The energy conversion efficiency can be improved by optimizing the geometrical parameters of the EMAT. The single-variable design method used to optimize the geometric parameters of the transducer cannot solve the multi-variable problem. However, the multi-parameter design optimization of an EMAT involves a lot of computation. Therefore, in the optimization design of EMATs, the optimal design method with the highest possible optimization efficiency should be selected. The orthogonal test is a multi-factor and multi-level optimization design method based on mathematical and statistical theory. The selection is made according to the quantity and level of various factors in the test, and representative points are selected from the comprehensive test to achieve results comparable with those of a large number of full-scale tests with the least number of tests [25]. Zhang [26] et al. studied the influence of geometric parameters on the EMAT of a butterfly coil using orthogonal tests and designed an improved butterfly coil EMAT, whose signal was 4.97 times higher than that of the original butterfly coil EMAT. Jia et al. [27] studied the degree of influence of the main parameters of the transducer on signal amplitude, and axial and radial focus migration using an orthogonal test method and selected the optimal parameter combination. Thus, the focusing performance of the PFSV-EMAT is improved. Vladimir et al. [28] studied the problem of robust discrimination of ARX models with constrained output variance using the optimal input design. By iterating an adaptive process between parameter estimation and input design using the current parameter estimation, the optimal input further increases the convergence speed of the robust algorithm. Compared with this design, the design process of the orthogonal test is simple and the number of tests is fewer.

Researchers have previously studied the influence of vertically polarized permanent magnets on single-mode Lamb waves and have enhanced the acoustic amplitude merely by improving the intermediate physical fields, including the eddy current, magnetic fields, or Lorentz force fields of the transmission processes of EMATs. To date, however, little work has been carried out on horizontally polarized permanent magnets by investigating and comparing the influence of various EMAT parameters on the amplitude. In addition, since the wavelength of the A0 wave is smaller than that of the S0 wave, the A0 wave is more sensitive to defects than the S0 wave [29].

Therefore, in this paper, an HP-PPM-EMAT is proposed to enhance the static magnetic field in the horizontal direction and to generate A0 waves. The unique feature of this design is that most of the horizontal component of the magnetic field is concentrated around the center wire, giving full play to the role of the magnet, which can not only enhance the A0 wave but also significantly eliminate the excess S0 wave. For a Lamb wave EMAT, the HP-PPM-EMAT solves the problem of a low energy conversion efficiency of EMATs and the multi-mode challenge of the Lamb wave. Compared with previous optimizations of the EMAT, this work adopts two stages of optimization design, namely orthogonal tests of the main geometric parameters of the HP-PPM-EMAT and width optimization of part of the magnet elements of the HP-PPM-EMAT. The simulation results show that the improved HP-PPM-EMAT significantly improves the signal purity of A0 wave.

This paper is organized as follows. The configuration and working principle of the proposed HP-PPM-EMAT are described in Section 2. In Section 3, the finite element models (FEMs) of the traditional EMAT and the HP-PPM-EMAT are established, and the magnetic flux density, Lorentz force, and signal intensity of the traditional EMAT and the HP-PPM-EMAT are compared and analyzed. Simulation results show that the proposed HP-PPM-EMAT can significantly enhance the A0 mode and suppress the S0 mode. Subsequently, in Section 4, orthogonal tests are adopted to study the permanent magnet height, the coil lift-off distance, the coil width, and the coil height of the HP-PPM-EMAT, and the optimal parameter combination is found. On this basis, the width of magnet units is improved. The last section (Section 5) concludes through numerical simulation that, compared with traditional EMATs, the improved HP-PPM-EMAT can generate a purer A0 mode Lamb wave.

## 2. Configuration and Working Principle of EMATs

### 2.1. HP-PPM-EMAT

A traditional Lamb wave EMAT consists of a rectangular magnet, a metal specimen, and a meander coil, where the magnet is horizontally polarized, and a pulse excitation is applied inside the coil. The metal specimen is an aluminum plate, and its electromagnetic ultrasonic energy transduction mechanism is mainly Lorentz force [30], as shown in Figure 1a. The meander coil is a multi-fold structure coil, and it is easier to generate Lamb waves in the aluminum plate when combined with the rectangular magnet, as shown in Figure 1b. The traditional EMAT produces Lamb waves of two modes, at least in the metal samples, which is not conducive to thickness measurement and defect detection of metal plates. To surmount this issue, a novel EMAT is proposed, whose permanent magnet consists of eight magnet units arranged periodically, as shown in Figure 1b,c. The height of the rectangular magnet *h*_1_ is equal to that of the magnet unit *h*_2_, which is 5 mm. The rectangular magnet width *w*_1_ is 34 mm, equivalent to the sum of the width of the eight magnet units *w*_2_ and the width of the air domain between adjacent magnet units. The traditional EMAT coil spacing *d*_1_ is the same as the HP-PPM-EMAT coil spacing *d*_2_, wherein the coil spacing of the HP-PPM-EMAT determines the spacing *L* of adjacent magnet units. The height *t*_1_ and width *w*_3_ of the coil are 0.05 mm and 0.5 mm, respectively, and the distance *h*_3_ from the upper surface of the aluminum plate is 0.4 mm. Both configurations are loaded horizontally for consistency, and the residual magnetic flux density is 1.2 T, differing only in the magnet configuration. The size of the aluminum plate is 400 mm × 400 mm × 2 mm. The model parameters are shown in Table 1. The meander coil is made of a copper coil with printed circuit board (PCB) technology. The pulse current signal in the excitation coil is a cosine Hanning window signal with five cycles, as expressed by
(1)$$I=A_{0}\cdot \cos(2\pi \;ft)[1-\cos(2\pi \;ft\;/\;n)],(0<t<n\;T)$$
where *A*_0_ is the amplitude of the excitation signal and is set to 20A; *f* is the center frequency of the excitation signal, which is 190 kHz; *t* is the time variable; and *n* is the number of cycles. The excitation current signal is shown in Figure 2.

### 2.2. Working Principle of EMATs

A traditional EMAT and the HP-PPM-EMAT work on the same principle, as shown in Figure 3. The meander coil is located between the aluminum plate and the permanent magnet. After the exciting current ***J*_0_** is passed into the meander coil, a dynamic magnetic field is generated in the aluminum plate, and an eddy current ***J*_e_** is induced at the skin depth of the aluminum plate surface. The eddy current ***J*_e_** is expressed by
(2)$$J_{e}\;=-\sigma \frac{\partial A}{\partial t}$$
where ***A*** is the magnetic vector potential, σ is the conductivity, and *t* is the time. A static magnetic field ***B*_s_** provided by the periodic permanent magnets and the dynamic magnetic field ***B*_d_** are given by
(3)$$B_{d}=\nabla \times A$$
(4)$$B_{s}=\mu_{1}H+B_{r}$$
where $\mu_{1}$ is the permeability, ***H*** is the magnetic field intensity, and ***B*_r_** is the remanent magnetic flux density. The static magnetic field Lorentz force ***F*_s_** and the dynamic magnetic field Lorentz force ***F*_d_** promote the specimen particles under the wire to vibrate at high frequency, thereby generating ultrasonic waves. The total Lorentz force is
(5)$$F_{L}=F_{s}+F_{d}=J_{e}\times B_{s}+J_{e}\times B_{d}$$

It is worth noting that due to the instability and uncontrollability of the dynamic magnetic field, the intensity of the dynamic magnetic field is usually weakened and the intensity of the static magnetic field is increased to make ***F*_L_** ≈ ***F*_s_** [31].

Figure 4 shows the waveform structure of the Lamb wave on a 2 mm thick aluminum plate with an excitation frequency of 190 kHz. U_x_ represents the in-plane displacement along the direction of wave propagation; U_y_ represents the out-of-plane displacement, perpendicular to the direction of wave propagation. Figure 4a shows the wave structure of the S0 wave, where U_x_ is maximum at the center of the plate thickness and decreases from the middle to the upper and lower edges of the plate; U_y_ is at its minimum, zero, at the center of the plate thickness, and gradually increases from the center to both sides. As can be seen from the normalized displacement, U_x_ is more excellent than U_y_, indicating that the S0 mode is dominated by in-plane displacement. Figure 4b shows the wave structure of the A0 wave, where U_x_ is at its minimum, zero, at the center of plate thickness, and gradually increases from the center to both sides; U_y_ reaches its maximum at the center of plate thickness and decreases from the middle to the upper and lower edges of the plate. According to the normalized displacement, U_x_ is less than U_y_, indicating that the A0 mode is dominated by out-of-plane displacement. Therefore, to enhance the A0 mode and to suppress the S0 mode, boosting U_y_ and inhibiting U_x_ are necessary. However, the displacement depends on the size of Lorentz force. According to Equation (4), the magnitude of Lorentz force is determined via the static magnetic field *B_s_*, the vertical magnetic field produces a horizontal Lorentz force, and the horizontal magnetic field produces a vertical Lorentz force. Strengthening the vertical magnetic field component increases the horizontal Lorentz force and weakening the horizontal magnetic field component decreases the vertical Lorentz force, ultimately strengthening the A0 wave and suppressing the S0 wave.

The dispersion characteristic curve of the low-order Lamb wave in an aluminum plate is shown in Figure 5. The phase velocity dispersion curve can be used to determine the excitation condition of the desired mode, and the group velocity dispersion curve can be used to predict the wave propagation velocity of the excited mode. It can be seen from the dispersion curve that at least two modes are excited simultaneously in the aluminum plate, which increases the difficulty of signal analysis. Therefore, in order to reduce multiple modes, locations less than the cut-off frequency of A1 mode tend to be selected as the excitation frequency. The frequency of the chosen operating point in this paper is 190 kHz, and the thickness of the aluminum plate is 2 mm. According to Figure 5a,b, the phase velocity and group velocity of the S0 mode and A0 mode are 5455 m/s and 5430 m/s, and 1709 m/s and 2753 m/s, respectively. The Lamb wave wavelength *λ* is
(6)$$\lambda =\frac{C_{p}}{f}$$
where *C_p_* is the phase velocity and *f* is the excitation frequency, so the wavelength is 9.0 mm. In order to realize the phase interference and to improve the direction and amplitude of the excited ultrasonic Lamb wave, the meander coil spacing is set to 4.5 mm, one-half wavelength.

## 3. FEM and Simulation Analysis of EMATs

### 3.1. Established the FEMs

Two-dimensional FEMs of the traditional EMAT and the HP-PPM-EMAT were established in COMSOL Multiphysics 6.0, as shown in Figure 6. Two observation points, P1 and P2, located on the right side of the aluminum plate were set, where P2 was further away from the EMAT than P1, and their coordinates were (260, 0.8) and (310, 0.8), respectively. The red line L1 on the surface of the aluminum plate started from the left of the magnet to the right of the magnet, and the *x*-axis ranged from −30 mm to 30 mm.

Figure 7 shows the meshes of the HP-PPM-EMAT model. The degree of mesh generation directly determines the calculation accuracy of the FEM. The more detailed the mesh section, the higher the calculation accuracy and the longer the solution time. Therefore, the simulation time should be shortened by as much as possible to ensure high precision, and relevant calculations and selective refinement of grid generation should be carried out. In this work, the mesh refinement is performed on the skin depth region of the aluminum plate surface, where the electromagnetic energy is concentrated and changes drastically during EMAT energy conversion. The skin depth of the aluminum plate *θ* is [32]
(7)$$\theta =\sqrt{\frac{\rho_{m}}{\pi \mu_{0}\mu_{m}f}}$$
where $\rho_{m}$ is the resistivity of the aluminum plate, $\rho_{m}$ = 2.62 × 10^−8^ Ω/m, $\mu_{0}$ is the vacuum permeability, $\mu_{0}$ = 4π × 10^−7^ H/m, $\mu_{m}$ is the relative permeability of the aluminum plate, $\mu_{m}$ = 1, *f* is the excitation frequency, and *f* = 190 kHz. According to Equation (7), the skin depth of the surface of the aluminum plate is 0.186 mm. When the maximum mesh size is less than λ/10, the finite element results converge, and the simulation results are accurate and reliable [33]. The maximum cell sizes of the grid for the permanent magnet, the coil, and the skin depth within the aluminum plate were set to 0.9 mm, 0.2 mm, and 0.1 mm, respectively. The induced current in the aluminum plate mainly exists in the skin depth, which is also the area where electromagnetic coupling occurs intensively. Therefore, the boundary layer was adopted on the upper surface of the aluminum plate. When the boundary layer is greater than the skin depth, the skin depth can be fully resolved, so the boundary layer was eight and the stretch factor was 1.2. The thickness of the first layer was 0.005 mm. Free split triangular meshes were used in other regions of the aluminum plate, and the maximum mesh size was set to 0.2 mm.

### 3.2. Analysis of Static Magnetic Field

Figure 8 shows the magnetic flux density distributions of the two magnets on line L1. As shown in Figure 8a, Bsx, the horizontal component of the magnetic flux density of the rectangular magnet, forms a flat peak below the magnet, which is significantly reduced at both ends of the magnet due to the edge effect; Bsy, the vertical component of the flux density, is almost zero at the center of the magnet, and there are two peaks of equal magnitude and opposite directions at both ends of the magnet. The peak sizes of Bsx and Bsy are 0.217 T and 0.435 T, respectively. It can be seen from Figure 8b that the magnetic flux density of the HP-PPM-EMAT approximately presents a periodic distribution. The peak sizes of Bsx and Bsy are 0.273 T and 0.339 T, respectively. Bsx has a peak in the center of each small magnet width; Bsy is approximately zero at the center of the width of the six small magnets. Despite the fact that Bsy is not zero on both sides, Bsx is greater than Bsy. Therefore, all wires of the HP-PPM-EMAT configuration should be arranged directly under the small magnet so as to achieve the maximum Bsx and minimum Bsy in the magnetic field environment of the coil. Furthermore, the horizontal component of the magnetic field produces the most significant Lorentz force in the vertical direction, and the horizontal Lorentz force generated by the vertical magnetic field component is the smallest, so the high-purity A0 wave is finally excited.

### 3.3. Analysis of Lorentz Force Field

The distributions of Lorentz force in line L1 illustrated in Figure 6 are shown in Figure 9. The components of the Lorentz force in the x and y directions are denoted by f_Lx_ and f_Ly_, respectively. By comparing Figure 9a’s traditional EMAT and Figure 9b’s HP-PPM-EMAT, it can be seen that the distribution of the f_Ly_ of the two EAMTs is similar, and there is regularity. There are significant differences in the distribution of the f_Lx_, which is manifested, on the one hand, by the multiple small peaks of the f_Lx_ in the HP-PPM-EMAT. On the other hand, the maximum value of the f_Lx_ in the traditional EMAT is 2.77 × 10^6^ N/m^3^, whereas the maximum value of the f_Lx_ in the HP-PPM-EMAT is 1.21 × 10^6^ N/m^3^, reduced by 56.3%. Meanwhile, the peak value of the f_Ly_ of HP-PPM-EMAT increased from 2.15 × 10^6^ N/m^3^ to 2.76 × 10^6^ N/m^3^, increasing by 28.4%. Here, the magnetic flux density distribution is combined for explanation, as shown in Figure 8. The horizontal magnetic field is mainly generated under the permanent magnet, and the vertical magnetic field is mainly generated on both sides of the permanent magnet. The magnetic flux density distribution in x and y components is different, leading to the difference in Lorentz force distribution. The x component of magnetic flux density in the traditional EMAT is distributed uniformly, and the y component has a large peak on both sides of the permanent magnet, which also causes the f_Lx_ to be distributed significantly on both sides and weakly in the middle. The HP-PPM-EMAT comprises multiple magnet units, and there is a magnetic field y component on both sides of each magnet unit. A small amount of Lorentz force is generated from the interaction between the magnetic field of the y component and the induced eddy current, which is manifested as multiple small peaks in the distribution of the f_Lx_. The induced eddy current presents a periodic distribution on the surface of the aluminum plate, and the f_Ly_ changes in a stable range, making the f_Ly_ distribution show a regular distribution. The HP-PPM-EMAT coil is located directly below the magnet units, and the x component of the magnetic flux density directly below all exceeds that of the traditional EMAT, so the peak value of the f_Ly_ is increased.

### 3.4. Analysis of Signal of EMATs

The displacement distribution is the probability that a molecule will shift in a specific direction and at a particular distance. Figure 10 shows the displacement distribution of traditional EMAT and HP-PPM-EMAT at *t* = 40 μs, 50 μs, and 60 μs, demonstrating the Lamb wave propagation process to the right over time, which was simulated using COMSOL Multiphysics 6.0. It can be seen from Figure 10 that there are two modes of Lamb wave. Since the velocity of the S0 wave is about twice that of the A0 wave, it is determined that the left side of Figure 10a,b is the A0 wave, and the right side is the S0 wave. In Figure 10a, the A0 mode displacement amplitude of the traditional EMAT is more significant than that of the S0 mode, and the two waves gradually propagate to the right with the increase in time. The A0 mode displacement amplitude of HP-PM-EMAT in Figure 10b is much greater than the S0 mode displacement amplitude. The propagation process of the A0 wave can be observed over time. On the other hand, the propagation process of the S0 wave is not significant. Compared with the traditional EMAT, the HP-PPM-EMAT can enhance the A0 wave and weaken the S0 wave simultaneously to obtain the Lamb wave with a relatively single mode. The displacement amplitude mentioned above refers to the maximum distance of the vibrating object from the equilibrium position, and the amplitude is numerically equal to the size of the maximum displacement.

Figure 11 shows the total displacement of the traditional EMAT and the HP-PPM-EMAT at P1, and Figure 12 shows the y-component displacement of the traditional EMAT and the HP-PPM-EMAT at P2, respectively. No matter the total or y-component displacement, the displacement of the Lamb wave has two wave packets. The distance between P1 and P2 is 50 mm. The propagation velocity of the two wave packets can be calculated according to the propagation distance and time difference. The propagation velocity of the first wave packet is 5495 m/s, close to the theoretical group velocity of 5430 m/s of the S0 mode at the excitation frequency of 190 kHz, and the relative error of the two group velocities is only 1.2%. When the excitation frequency is 190 kHz, the theoretical group velocity of A0 is 2753 m/s, and the propagation velocity of the second wave packet is 2683 m/s. The relative error of the two group velocities is only 2.5%. Therefore, the first wave packet in Figure 11 and Figure 12 can be determined to be the S0 wave and the second wave packet can be determined to be the A0 wave. It should be noted that the propagation velocities of S0 and A0 are calculated from the total displacement of the two points P1 and P2. According to Figure 11 and Figure 12, it can be calculated that the total displacement amplitude ratio of A0 to S0 generated via the traditional EMAT is 4.06, and the total displacement amplitude ratio of A0 to S0 generated via the HP-PM-EMAT is 16.53. Compared with the traditional EMAT, the A0 total displacement amplitude of the HP-PPM-EMAT increased by 19.8%, the S0 total displacement amplitude decreased by 240.0%, and the ratio of A0 to S0 total displacement amplitude increased by 3.07 times, significantly improving the signal purity of the A0 mode.

## 4. Optimization of EMATs

### 4.1. Orthogonal Test Design

In the parameters of the HP-PPM-EMA, the permanent magnet height *h*_2_, the coil lift-off distance *h*_3_, the coil width *w*_3_, the coil height *t*_1_, and the distance between the permanent magnet and aluminum plate have essential effects on the strength of Lamb wave signal, and these parameters directly affect the performance of EMATs. Since the distance between the permanent magnet and the tested piece will affect the stability of acoustic signal in actual detection, the lift-off distance of the permanent magnet is no longer used as a variable for analysis in parameter optimization [34].

The orthogonal test is a design method of multi-factor and multi-level research, which can not only efficiently find the optimal parameter combination of the HP-PPM-EMAT but also analyze the influence rule of each parameter on signal strength. In this work, a four-factor four-level orthogonal test was designed. The parameters *h*_2_, *h*_3_, *w*_3_, and *t*_1_ were defined as “factors”, and the values chosen for the parameters are called “levels”. The ranges of variation in the EMAT parameters, which are the factors of the array, were selected according to a group of common and realistic specifications, the emission power, and the local environment of the EMAT, which were *h*_2_ 4–24 mm, *h*_3_ 0.1–0.4 mm, *w*_3_ 0.25–1 mm, and *t*_1_ 0.03–0.15 mm. Table 2 shows the factors and levels of the orthogonal test. The total displacement amplitude ratio of A0 and S0 of the HP-PPM-EMAT was defined as the *δ*, and the symbol *δ* was used as the evaluation standard. The *δ* value directly reflected the purity of the A0 wave. The P1 was selected as the analysis point, and the total displacement amplitude ratio of A0 and S0 of particles obtained from 16 groups of orthogonal tables and simulation were summarized. The orthogonal test and simulation results of L_16_(4^4^) are shown in Table 3.

### 4.2. Analysis the Results of the Orthogonal Test Design

The arithmetic mean values *k*_1_, *k*_2_, *k*_3_, and *k*_4_ of the *δ* of four factors at the same level were calculated using the orthogonal test results. The magnitude of *k*_i_ represents the extent to which different levels of this factor affect the amplitude of Lamb wave displacement. The influence degree of each factor level on Lamb wave amplitude was determined by solving the range value *R* of *k*_i_. The greater the *R* value, the greater the factor’s influence on the excitation Lamb wave displacement amplitude. Table 4 shows the summary of orthogonal test results of the HP-PPM-EMAT, and the order of influence degree of each parameter on the *δ* is *h*_2_ (5.19) > *h*_3_ (1.96) *w*_3_ (1.22) > *t*_1_ (0.79). Figure 13 shows each factor’s average value and influence degree on the *δ*. The permanent magnet height *h*_2_ has the most significant effect on the *δ*. With the increase in *h*_2_, the *δ* decreases first and then increases. When *h*_2_ grows from 10 mm to 16 mm, the downward trend becomes gentle; when *h*_2_ is larger than 16 mm, the *δ* increases. The coil lift-off distance *h*_3_, the coil width *w*_3_, and the coil height *t*_1_ have the same effect on the *δ*, that is, with the decrease in the values of the three parameters, the *δ* shows an increasing trend. The *w*_3_ has little effect on the *δ*, while *t*_1_ has no significant impact on the *δ*. When *h*_2_ = 4 mm, *h*_3_ = 0.1 mm, *w*_3_ = 0.25 mm, and *t*_1_ = 0.03 mm, the *δ* value is the largest. Therefore, to improve the signal purity of the A0 wave, the optimal parameters of the HP-PPM-EMAT design model are *h*_2_ = 4 mm, *h*_3_ = 0.1 mm, *w*_3_ = 0.25 mm, and *t*_1_ = 0.03 mm. The *δ* value optimized with the orthogonal test is 1.31 times that of the HP-PPM-EMAT before optimization and 5.49 times that of the traditional EMAT.

### 4.3. Improvement in the Width of Magnet Units

This work determined the coil spacing at the excitation frequency of 190 kHz, and the spacing of adjacent magnets and the magnet width were defined under the condition that the coil spacing is 4.5 mm. The coil spacing of the HP-PM-EMAT was determined with the A0 wave wavelength corresponding to the selected working point. Considering that the total width of the permanent magnet and the coil spacing remain unchanged and that it is also necessary to keep the coil directly below each permanent magnet unit, it is impossible to change the width of each magnet unit. Therefore, the orthogonal test did not involve the width of the magnet units. However, as another important parameter of the HP-PPM-EMAT, the influence of the width of magnet units on the *δ* value is unknown. Here, the width of the magnet units was improved separately based on the HP-PPM-EMAT optimized with orthogonal tests. Each small magnet was numbered in order from left to right as 1–8. The width of magnet No. 1 and No. 8 remained unchanged, and the group number and serial number of magnets are shown in Table 5.

The widths of magnet No. 2, No. 3, No. 4, No. 5, No. 6, and No. 7 varied from 1.0 mm to 4.3 mm with a step size of 0.3 mm. Figure 14 shows the effect of the width of the magnet units on the *δ*, and the coordinates in the figure are the best points of each group. The *δ* of the fourth group increased with the increase in magnet width, while the other five groups increased first and then decreased. The optimal values of group 1, group 3, group 5, and group 6 are close to 23, which is close to the *δ* of the HP-PPM-EMAT after the orthogonal test. The optimal values of group 2 and group 4 are 28.0 and 35.5, respectively. For group 2, the *δ* value of the EMAT under this configuration is 1.65 times that of the HP-PPM-EMAT when the width of magnet No. 4 and No. 5 are *w*_4_ = 1.3 mm. For group 4, the *δ* value of the EMAT under this configuration is 2.09 times that of the HP-PPM-EMAT and 8.74 times that of the traditional EMAT when the widths of magnet No. 3, No. 4, No. 5, and No. 6 are *w*_5_ = 4.3 mm. Therefore, the improved EMAT configuration can excite the A0 wave with a higher purity than the HP-PPM-EMAT configuration. The HP-PPM-EMAT of group 2 and group 4 are shown in Figure 15.

To further prove that the improved EMAT can excite A0 waves of higher purity, Figure 16 shows the amplitudes of S0 mode signals of the improved EMAT, the HP-PPM-EMAT, and the traditional EMAT. Figure 16a shows the x-component displacement of the S0 mode. The S0 displacement amplitudes of the three EMATs in the x-component are 6.36 × 10^−9^, 1.46 × 10^−8^ mm, and 5.10 × 10^−8^ mm, respectively. The S0 displacement amplitude of the x-component of the improved EMAT is reduced by 1.30 times compared with the HP-PPM-EMAT and 7.01 times compared with the traditional EMAT. Figure 16b shows the y-component displacement of the S0 mode. The S0 displacement amplitudes of the three EMATs in the y-component are 5.20 × 10^−10^ mm, 1.18 × 10^−9^ mm, and 4.26 × 10^−9^ mm, respectively. The S0 displacement amplitude of the y-component of the improved EMAT is reduced by 1.27 times compared with the HP-PPM-EMAT and 7.20 times compared with the traditional EMAT. This means that the improved EMAT can produce a single A0 wave when a reasonable width of the magnet units is selected.

## 5. Conclusions

The static magnetic field of a traditional A0 wave EMAT is provided via a single rectangular permanent magnet, which produces a poor purity of the A0 mode Lamb wave. To suppress the S0 mode and to enhance the A0 mode, an HP-PPM-EMAT configuration is proposed, which improves the magnetic flux density at local locations. Through simulation, the traditional EMAT and the HP-PPM-EMAT are compared and analyzed. According to the simulation results of static magnetic field, Lorentz force, and displacement signal, compared with the traditional EMAT, the HP-PPM-EMAT has an enhanced effect on the A0 mode and the suppressed S0 mode. Then, the magnet height *h*_2_, the coil-lifting distance *h*_3_, the coil width *w*_3_, and the coil height *t*_1_ of HP-PPM-EMAT were optimized using the orthogonal test. In terms of increasing the total displacement amplitude ratio of A0 to S0, *h*_2_ has the greatest effect on the *δ*. When *h*_2_ = 4 mm, *h*_3_ = 0.1 mm, *w*_3_ = 0.25 mm, and *t*_1_ = 0.03 mm, the *δ* value is the largest, which is 1.31 times that before the orthogonal test.

Considering that the total width of the permanent magnet and the distance between the coil are constant, the width of the middle six magnet units is optimized while the width of the magnet units on both sides of the periodic magnet is unchanged. The simulation results show that when the width of magnet No. 3, No. 4, No. 5, and No. 6 *w*_5_ = 4.3 mm and the width of the other four magnet units remained unchanged, the *δ* value reached the maximum, which is 2.09 times that of the HP-PPM-EMAT and 8.74 times that of the traditional EMAT. The improved EMAT generates a Lamb wave with a higher purity A0 mode, effectively solving the Lamb wave multi-mode problem. In the future, the omnidirectional acoustic Lamb wave EMAT will be developed based on the PPM EMAT proposed in this paper to realize the detection of aircraft rotor ice thickness and plate structure defects.

## Figures and Tables

**Figure 1 sensors-23-07117-f001:**
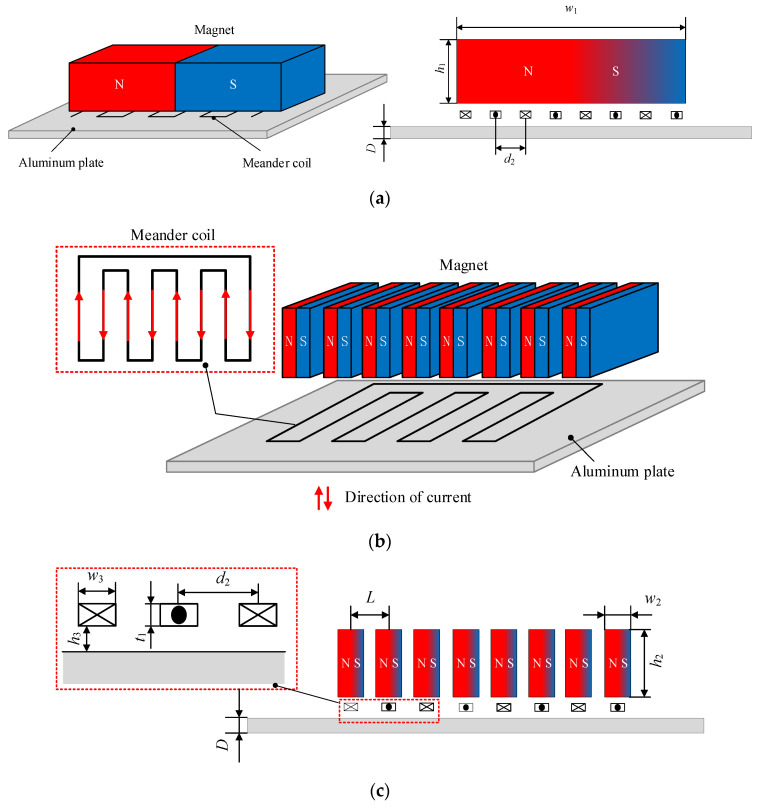
Configuration of EMATs: (**a**) the traditional EMAT; (**b**) 3D model of the HP-PPM-EMAT; (**c**) 2D model of the HP-PPM-EMAT.

**Figure 2 sensors-23-07117-f002:**
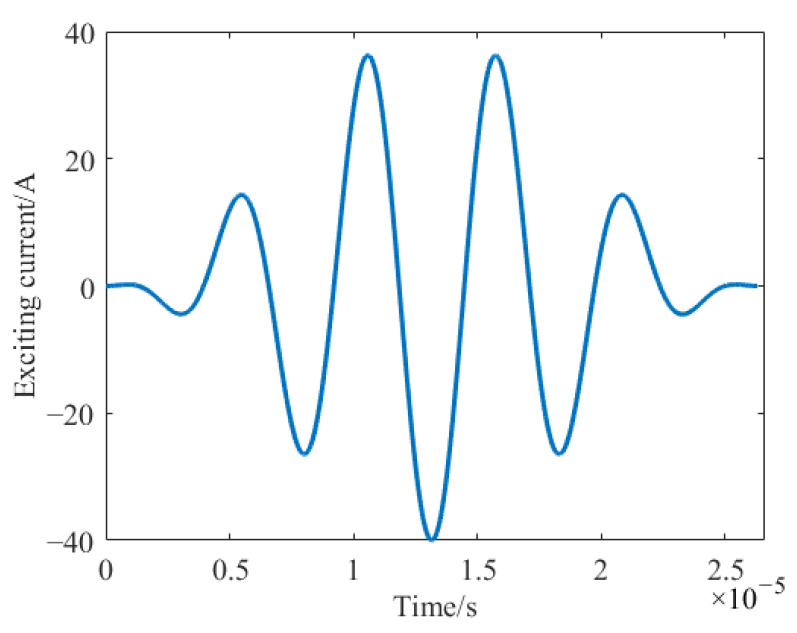
Five-cycle cosine signal modified using a Hanning window with a center frequency of 190 kHz.

**Figure 3 sensors-23-07117-f003:**
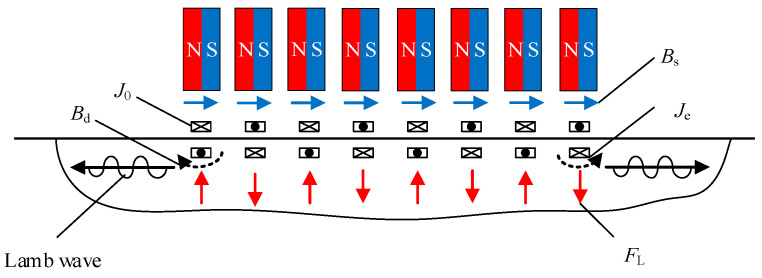
Schematic diagram of the working principles of the HP-PPM-EMAT.

**Figure 4 sensors-23-07117-f004:**
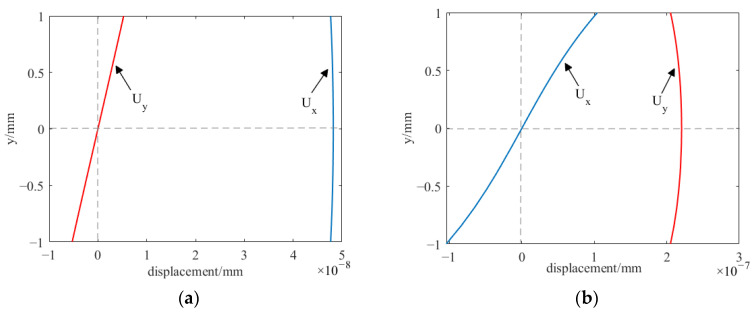
Wave structures of ultrasonic-guided waves at 190 kHz on a 2 mm thickness aluminum plate. (**a**) S0 mode; (**b**) A0 mode.

**Figure 5 sensors-23-07117-f005:**
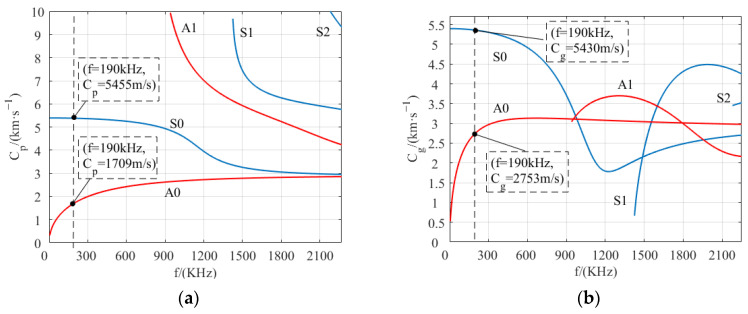
Dispersion curves of ultrasonic Lamb waves in a 2 mm thick aluminum plate. (**a**) Phase velocity; (**b**) Group velocity.

**Figure 6 sensors-23-07117-f006:**
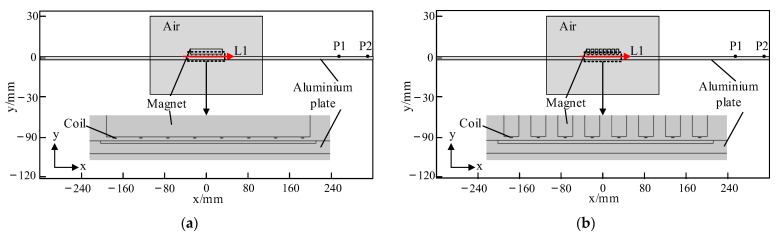
Finite element models of traditional EMAT and HP-PPM-EMAT. (**a**) Traditional EMAT, P1(260, 0.8) and P2(310, 0.8). (**b**) HP-PPM-EMAT, P1(260, 0.8) and P2(310, 0.8).

**Figure 7 sensors-23-07117-f007:**
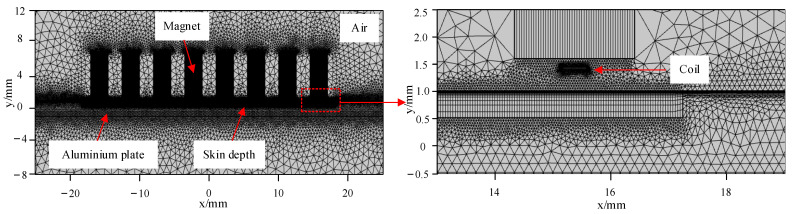
Meshes of the HP-PPM-EMAT model.

**Figure 8 sensors-23-07117-f008:**
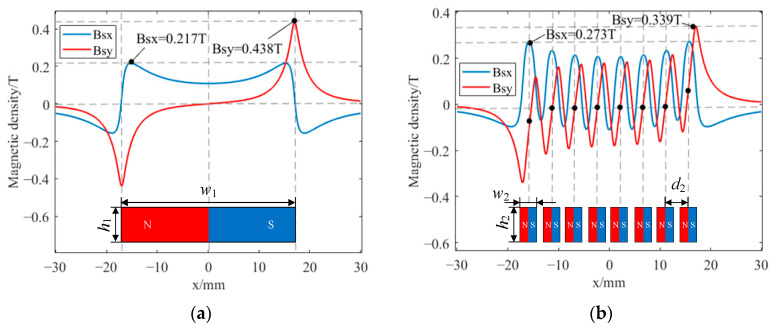
Distribution of magnetic flux density on line L1 for (**a**) traditional EMAT and (**b**) HP-PPM-EMAT.

**Figure 9 sensors-23-07117-f009:**
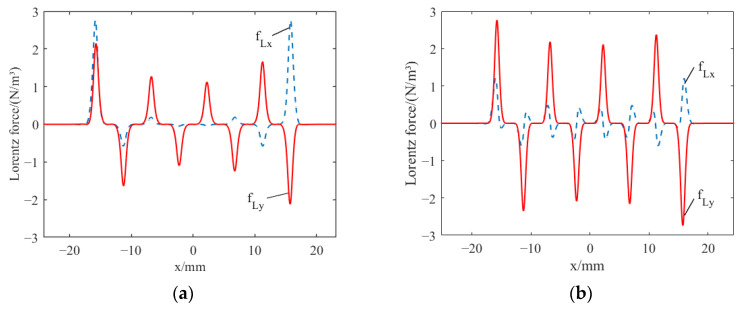
Distribution of Lorentz force on line L1 at *t* = 6.3 μs for (**a**) traditional EMAT and (**b**) HP-PPM-EMAT.

**Figure 10 sensors-23-07117-f010:**
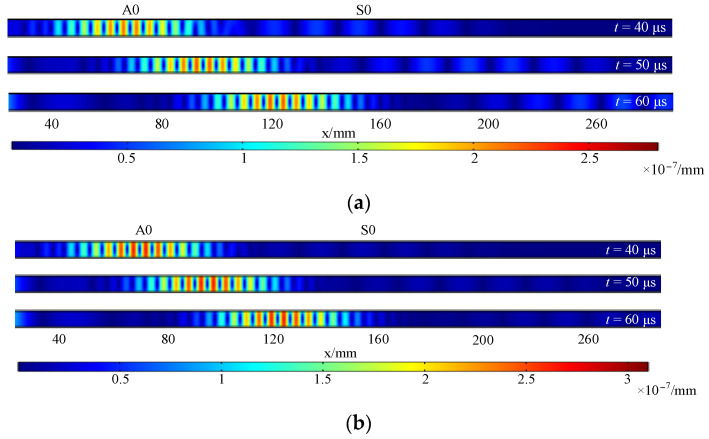
Displacement distribution of (**a**) traditional EMAT and (**b**) HP-PPM-EMAT.

**Figure 11 sensors-23-07117-f011:**
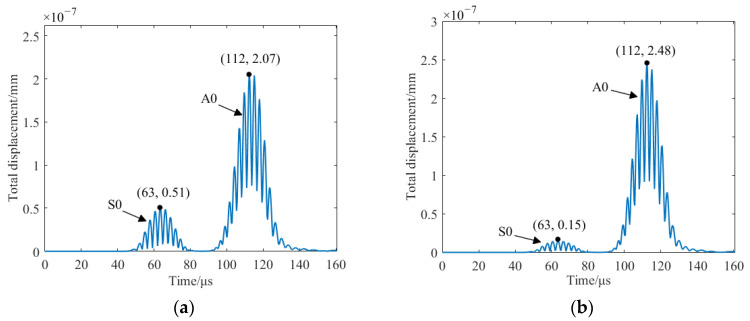
Total displacement on the surface of the aluminum plate at P1 to (**a**) traditional EMAT and (**b**) HP-PPM-EMAT.

**Figure 12 sensors-23-07117-f012:**
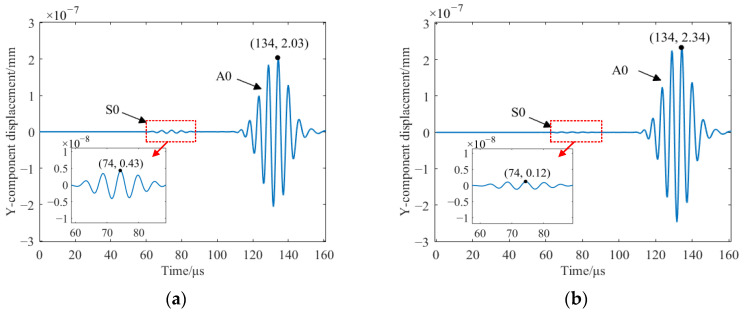
Y-component displacement on the surface of the aluminum plate at P2 to (**a**) traditional EMAT and (**b**) HP-PPM-EMAT.

**Figure 13 sensors-23-07117-f013:**
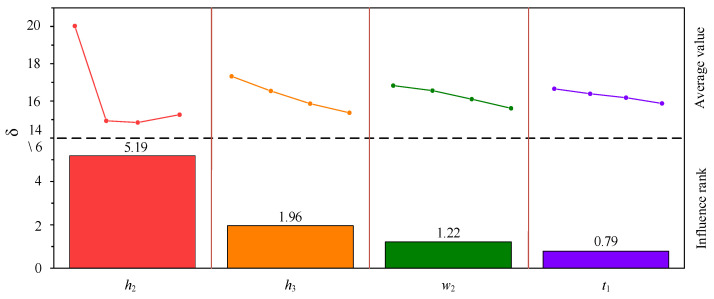
Range analysis of the results for the orthogonal test design; the column height represents the average value of each level and the influence rank for each factor.

**Figure 14 sensors-23-07117-f014:**
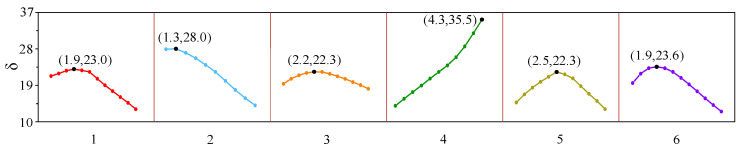
Influence of magnet unit width on *δ*.

**Figure 15 sensors-23-07117-f015:**
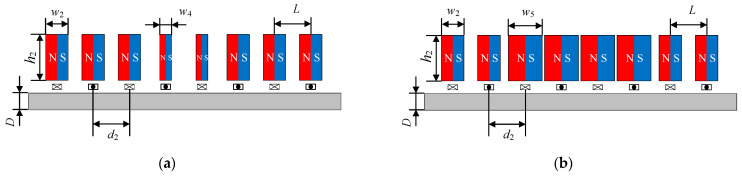
Structure diagram of (**a**) group 2 and (**b**) group 4.

**Figure 16 sensors-23-07117-f016:**
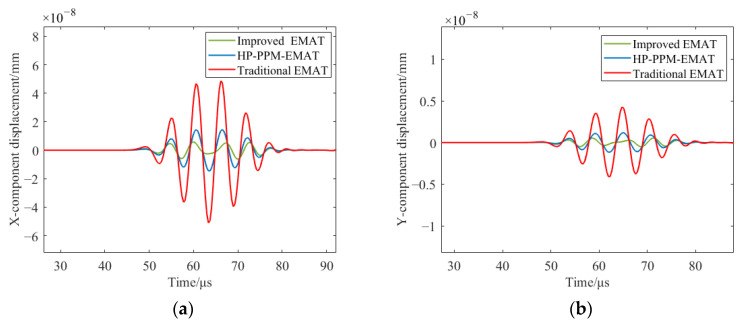
Displacement on the surface of the aluminum plate at P1 to (**a**) x-component displacement and (**b**) y-component displacement.

**Table 1 sensors-23-07117-t001:** Parameters of 2-D FEM used in this work.

Model	Parameters	Symbols	Value
	Width	*w* _2_	2.5 mm
	Height	*h* _2_	5 mm
Magnet	Type	N35	1.2 T
	Quantity		8
	Spacing of magnet	*L*	4.5 mm
	Spacing	*d*_1_, *d*_2_	4.5 mm
	Width	*w* _3_	0.5 mm
	Height	*t* _1_	0.05 mm
Coil	Lift-off distance	*h* _3_	0.4 mm
	Relative permeability	*μ* _r_	1
	Conductivity	σ	5.998 × 10^7^ S/m
	Relative dielectric constant	*ε*	1
	Thickness	*D*	2 mm
	Mass density	*ρ*	2700 kg/m^3^
Aluminum	Electrical conductivity	*σ_A_*	3.77 × 10^7^ S/m
	Young’s modulus	*E*	70 × 10^9^ Pa
	Passion’s ratio	*μ*	0.33

**Table 2 sensors-23-07117-t002:** Factors and levels for the orthogonal test.

Factor	*h*_2_ (mm)	*h*_3_ (mm)	*w*_3_ (mm)	*t*_1_ (mm)
Level 1	4	0.1	0.25	0.03
Level 2	10	0.2	0.5	0.07
Level 3	16	0.3	0.75	0.11
Level 4	24	0.4	1	0.15

**Table 3 sensors-23-07117-t003:** L_16_(4^4^) orthogonal test table for the HP-PPM-EMAT.

Factor	*h*_2_ (mm)	*h*_3_ (mm)	*w*_3_ (mm)	*t*_1_ (mm)	*δ*
1	4	0.1	0.25	0.03	22.30
2	4	0.2	0.5	0.07	20.75
3	4	0.3	0.75	0.11	19.24
4	4	0.4	1	0.15	17.82
5	10	0.1	0.5	0.11	15.93
6	10	0.2	0.25	0.15	15.24
7	10	0.3	1	0.03	14.30
8	10	0.4	0.75	0.07	14.23
9	16	0.1	0.75	0.15	15.39
10	16	0.2	1	0.11	14.62
11	16	0.3	0.25	0.07	14.85
12	16	0.4	0.5	0.03	14.51
13	24	0.1	1	0.07	15.67
14	24	0.2	0.75	0.03	15.50
15	24	0.3	0.5	0.15	15.00
16	24	0.4	0.25	0.11	14.87

**Table 4 sensors-23-07117-t004:** Summary of orthogonal test results for the HP-PPM-EMAT.

Results	Level	Factors			
		*h*_2_ (mm)	*h*_3_ (mm)	*w*_3_ (mm)	*t*_1_ (mm)
*δ*	*k* _1_	20.03	17.32	16.82	16.65
	*k* _2_	14.93	16.53	16.55	16.38
	*k* _3_	14.84	15.85	16.09	16.17
	*k* _4_	15.26	15.36	15.60	15.86
	*R*	5.19	1.96	1.22	0.79
	Influence rank	*h*_2_ (5.19) > *h*_3_ (1.96) *w*_2_ (1.22) > *t*_1_ (0.79)			
	Preferred value	*h*_2_: 4 mm; *h*_3_: 0.1 mm; *w*_2_: 0.25 mm; *t*_1_: 0.03 mm;			

**Table 5 sensors-23-07117-t005:** Magnet group number and serial number.

Group Number	Serial Number
1	4
2	4, 5
3	3, 4, 5
4	3, 4, 5, 6
5	2, 3, 4, 5, 6
6	2, 3, 4, 5, 6, 7

## Data Availability

Not applicable.

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
