# Peer review of "Antisymmetric Lamb Wave Simulation Study Based on Electromagnetic Acoustic Transducer with Periodic Permanent Magnets"

_sensors, 2023, doi:10.3390/s23167117_

Round 1
Reviewer 1 Report
In this paper, a HP-PPM-EMAT is proposed, analyzed and optimized. There are several issues suggested for revision.
1. The originality of the manuscript is not obvious. It is suggested that the author highlight the contribution of the article.
2. The necessary experimental studies are lacking.
3. It is suggested to give more comparative analysis before and after optimization.
4. The conclusions were suggested to be further simplified.
5. The electromagnetic parameters of the coil need to be added in Table 1.
6. In Figure 8, Figure 9 and Figure 10, the names of figures (a) and (b) need to be described separately.
7. How the parameters of traditional EMAT and HP-PPM-EMAT are determined. These preliminary parameters also need to be given.
Reviewer 2 Report
There are aspects that are worthy of publication, however, this paper requires a major attention and careful considerations. Adequate revisions to the following points should be undertaken in order to justify recommendation for publication.
1. The advantages of the proposed method of this paper should be more highlighted.
2. The quality of language could be improved, for the benefit of journal readers.
3. New ideas with unique features compared to existing/published papers should be more highlighted. The topic is over-published, and it is difficult to identify the novelty with respect to existing literature, some of which has not been discussed despite its relevance. A quick search gives: Robust Kalman filtering for nonlinear multivariable stochastic systems in the presence of non-Gaussian noise, International Journal of Robust and Nonlinear Control; Optimal experiment design for identification of ARX models with constrained output in non-Gaussian noise, Applied Mathematical Modelling; It could be the object of a brief consideration focused on the advances on the topic and make relation with this paper, which could be discussed in Introduction section in the context of a more comprehensive literature review.
4. Authors should argue their choice of the performance evaluation indicators.
5. The conclusions should be extended and future lines of research should be discussed with more care.
6. The experimental setup is well discussed, but the authors are encouraged to further discuss the results in more technical details.
The quality of language could be improved, for the benefit of journal readers.
Reviewer 3 Report
1. The work is by FEM simulation. The title should be clear about it by adding "Simulation study...".
2. The work is performed by a 2-D finite element model. Can the authors estimate how deep is the device for the assumption to stand.
3. The Meander coil composition is not clear. PCB is printed circuit board? The electric properties of the polymeric insulating material should be considered in the simulation to preclude the capacitive effect. The current should run through the coil, however, Direction of the current shown in Fig. 1 is not so.
4. It is good that the authors did a simple procedure of optimization (Range analysis). I found that the h3, w2, and t1 did not reach the critical points. Supposedly, δ is decreasing with h3, w2, and t1?
5. The The presentation of Fig.4 is misleading. Both thickness/mm and normalized amplitude are not standard notation. The same problem can be found in every X-Y graph.
6. The style of bibliography is not consistent and maybe the number of references can be raised to 30.
There is no significant grammar errors, however the English is not good enough. The text should be readable smoothly.
Reviewer 4 Report
1. When describing the configuration of EMAT and HP-PPM-EMAT, the term "meander coil" is used. From the description and Figure 1, the reviewer understands that this is just a bent wire through which current is passed. Perhaps the author uses stable terminology in his description, which is not clear to everyone, or the reviewer does not understand everything correctly. Need to explain what a meander coil is? What is meant by turns of a coil? From figure 1 it is not clear that there are six of them, as indicated in the text. The configuration of the “meander coil” or rein in figure 3 differs from figure 1. This circumstance requires clarification. What is a “meander coil”, how is it exactly located relative to the magnet and the plate?
2. Line 106: «The metal specimen is an aluminum plate, and its electromagnetic ultrasonic energy transduction mechanism is mainly Lorentz force [21], as shown in Figure 1(a)» Perhaps in Figure 1 one should indicate the direction of the Lorentz force.
3. Lines 122 - 124 say that "... cosine pulse signal is used for excitation current, the cycle number is 5, the amplitude is 20 A, and the excitation frequency is 190 kHz. The excitation current signal is shown in Figure 2". It would be reasonable to give the mathematical formula of the signal, which is obviously the sum of six harmonics. The figure shows that the current amplitude for the total signal exceeds 25 A, but the text (line 123) talks about 20 A.
4. The static magnetic field represented by formula (3) consists of two parts. It's not clear why? This formula needs more explanation. It does not have a magnetic constant and does not indicate the value of magnetic permeability. What are the physical grounds for presenting the magnetic field of a permanent magnet in this form?
5. Part 3 of the article describes the mathematical model of EMAT. However, this description is not complete. The authors discuss the geometry of the model and the computational grid, but nothing is said about the equations, boundary conditions, and values of the material parameters for which the calculation was performed.
6. The authors indicated that they wanted to reduce the time and maintain the accuracy of the study and provided the exact values of the grid parameters at which the calculation was performed. The question of how the grid parameters were chosen (number of boundary layers, stretch coefficient, max dimensions) remains unanswered.
7. In Figure 9, there are two fragments. What does each of these fragments correspond to? The text of Section 3.3, where Figure 9 is mentioned, does not mention this. Section 3.3 should be supplemented with information about fragments (a) and (b) of Figure 9.
8. It should be clarified what the authors mean by the terms "Displacement Field", "displacement distribution" and "displacement amplitude" in section 3.4.
9. It is necessary to explain how the results presented in Figures 10 and 11 were obtained. From the text of Section 3.4, it can be concluded that these results were probably obtained during the experiments. Is it so?
10. Section 4.1 presents an orthogonal design of a multivariate experiment, which is further analyzed. The value d appears in the plan as an output factor (result). It is not clear how this value was obtained, as a result of an experiment or a numerical calculation? If the results of an experiment are discussed in Sections 3.4, 4.1, and 4.3, then its conditions should be described in the paper and a diagram of the experimental setup should be given.
11. The choice of levels of factors of the orthogonal test should be explained. Why can't you choose other levels? For example, from Table 3 it follows that the maximum value of the parameter δ was at the minimum value of the factor h2. It may make sense to consider a magnet height of 3mm or less. And it will improve the performance of EMAT. What are the restrictions and rules when choosing input factors?
12. The writing of all variables in the text and formulas must be brought to a single form and, possibly, written in italics.
13. Authors need to carefully edit the list of cited literature, bring it to a unified form and indicate the full imprint of the sources.
Reviewer 5 Report
1. A horizontally polarized periodic permanent magnet electromagnetic acoustic transducer (HP-PPM-EMAT) was proposed based on the traditional EMAT. A 2-D finite element model was then developed to perform a comparative analysis of magnetic flux density, Lorentz force, and signal strength between the traditional EMAT and the HP-PPM-EMAT
2. In the figure 2, 5-cycle cosine signal modified by a Hanning window with a center frequency of 190 kHz should be demonstrated in detail.
3. In the figure 4, wave structures of ultrasonic guided waves at 190 kHz on a 2-mm-thickness aluminum plate should be demonstrated in detail.
4. In the figure 5, dispersion curves of ultrasonic Lamb waves in a 2-mm-thick aluminum plate should be demonstrated in detail.
5. In the figure 8, distribution of magnetic flux density on line L1 should be demonstrated in detail.
6. In the figure 12, Y-component displacement on the surface of the aluminum plate at P2 to(a) traditional 307 EMAT and (b) HP-PPM-EMAT, should be demonstrated in detail.
7. The authors are suggested to highlight the contributions of the proposed work, compared to the prior works. A detailed discussion about prior works are suggested to add.
8. Some future works are suggested to discuss to give an enlighten to the readers.
9. Revise the English thoroughly before re-submission.
Moderate editing of English language required.
Round 2
Reviewer 1 Report
The quality of the manuscript has been effectively improved.
Author Response
Thank you very much for your valuable suggestion.
Reviewer 2 Report
The authors have not yet justified the contributions of their paper. What is the actual improvement with respect to other existing related results? It is necessary to additionally improve the basis of the contribution in relation to more recent references. Some recent relevant contributions appear to have been missed. The literature review on the topic is not thorough, some related results, proposed in the previous review round, should be included in this paper, as well as: Optimal experiment design for identification of ARX models with constrained output in non-Gaussian noise, Applied Mathematical Modelling;, give a short comment and in that way, point out other approaches and possibilities. After this, the manuscript may be considered for publication.
Minor editing of English language required
Reviewer 5 Report
no further comment.
Author Response

(The authors gave the same response as above.)
